# Cytoplasmic and Nuclear Forms of Thyroid Hormone Receptor β1 Are Inversely Associated with Survival in Primary Breast Cancer

**DOI:** 10.3390/ijms21010330

**Published:** 2020-01-03

**Authors:** Wanting Shao, Christina Kuhn, Doris Mayr, Nina Ditsch, Magdalena Kailuweit, Verena Wolf, Nadia Harbeck, Sven Mahner, Udo Jeschke, Vincent Cavaillès, Sophie Sixou

**Affiliations:** 1Department of Obstetrics and Gynecology, Breast Center, University Hospital, LMU Munich, 80337 Munich, Germany; Wanting.Shao@med-uni-muenchen.de (W.S.); Christina.kuhn@med.uni-muenchen.de (C.K.); nina.ditsch@uk-augsburg.de (N.D.); magdalena.kailuweit@swmbrk.de (M.K.); verena-wolf@gmx.net (V.W.); nadia.harbeck@med.uni-muenchen.de (N.H.); sven.mahner@med.uni-muenchen.de (S.M.); sophie.sixou@inserm.fr (S.S.); 2Department of Pathology, LMU Munich, 80337 Munich, Germany; doris.mayr@med.uni-muenchen.de; 3Department of Obstetrics and Gynecology, University Hospital Augsburg, Stenglinstr. 2, 86156 Augsburg, Germany; 4IRCM—Institut de Recherche en Cancérologie de Montpellier, INSERM U1194, Université de Montpellier, Institut régional du Cancer de Montpellier, CNRS, 208 rue des Apothicaires, F-34298 Montpellier CEDEX 5, France; vincent.cavailles@inserm.fr; 5Faculté des Sciences Pharmaceutiques, Université Paul Sabatier Toulouse III, 31062 Toulouse CEDEX 9, France; 6Cholesterol Metabolism and Therapeutic Innovations, Cancer Research Center of Toulouse (CRCT), UMR 1037, Université de Toulouse, CNRS, Inserm, UPS, 31037 Toulouse, France

**Keywords:** thyroid hormone receptor beta 1, subcellular localization, overall survival, breast cancer

## Abstract

The aim of this study was to investigate the expression of thyroid hormone receptor β1 (THRβ1) by immunohistochemistry in breast cancer (BC) tissues and to correlate the results with clinico-biological parameters. In a well-characterized cohort of 274 primary BC patients, THRβ1 was widely expressed with a predominant nuclear location, although cytoplasmic staining was also frequently observed. Both nuclear and cytoplasmic THRβ1 were correlated with high-risk BC markers such as human epidermal growth factor receptor 2 (HER2), Ki67 (also known as MKI67), prominin-1 (CD133), and N-cadherin. Overall survival analysis demonstrated that cytoplasmic THRβ1 was correlated with favourable survival (*p* = 0.015), whereas nuclear THRβ1 had a statistically significant correlation with poor outcome (*p* = 0.038). Interestingly, in our cohort, nuclear and cytoplasmic THRβ1 appeared to be independent markers either for poor (*p* = 0.0004) or for good (*p* = 0.048) prognosis, respectively. Altogether, these data indicate that the subcellular expression of THRβ1 may play an important role in oncogenesis. Moreover, the expression of nuclear THRβ1 is a negative outcome marker, which may help to identify high-risk BC subgroups.

## 1. Introduction

Breast cancer (BC), the most frequent cause of cancer death worldwide [1], is highly heterogeneous, leading to great complexity for diagnosis and therapy selection [2,3]. So far, only few diagnostic markers are well recognized in invasive BC, including expression of the two nuclear receptors (NR), the estrogen receptor (ER) and progesterone receptor (PR), and overexpression of human epidermal growth factor receptor 2 (HER2). Although therapies targeting ER and HER2 (e.g., tamoxifen and trastuzumab) have been very successful, some tumors ultimately develop resistance to single or even combination therapies [4]. Thus, the identification of other biomarkers is essential for optimal and personalized BC management.

Links between BC and expression of other NR have already been outlined by our lab and others [5,6,7,8,9,10]. Thyroid hormone receptors (THR) are members of the NR superfamily that mediate the classical genomic actions of thyroid hormone (TH) signaling in numerous tissues and regulate important physiological and developmental processes [11,12]. THR primarily act as ligand-dependent transcription factors, after heterodimerization with retinoid X receptor (RXR). Various factors influence TH activity, including THR mutations, interactions with heterodimerization partners and coregulators, and expression of various THR subtypes and their related intracellular localization [13,14,15]. Indeed, rapid shuttling of various THR isoforms between the nucleus and cytoplasm has been described, and such dynamic transport pathways may be linked to specific TH signaling activities in nucleus, cytoplasm, or mitochondria [11,12]. These properties have even led to a new classification scheme with four TH signaling pathways; the canonical pathway, in which liganded THR binds directly to DNA (type 1), is tethered to chromatin-associated proteins (type 2) or functions without recruitment to chromatin either in the nucleus or cytoplasm (type 3). Finally, in the type 4 pathway, TH acts at the plasma membrane or in the cytoplasm without binding THR [16].

A significant association between thyroid hormone (TH) signaling and BC has already been demonstrated [17,18,19]. High TH levels are correlated with advanced clinical stages of BC [20]. A negative relationship between the presence of nuclear saturable high affinity binding sites of TH and lymph node (LN) status of BC patients has also been known for decades. These binding sites have been named thyroid hormone receptors (THRs) [21]. In a study performed at mRNA level in 116 breast samples, both THRα and THRβ mRNA levels were decreased in BC compared with normal tissues; yet, only THRβ expression, and not that of THRα, was negatively associated with histological grade [22]. Literature regarding the clinical significance of THR at the protein level is still limited. A recent study performed in 41 invasive BC tissues suggested that nuclear THRα is down-regulated during breast carcinogenesis [23]. Other studies highlighted the role of THRβ as a tumor suppressor in BC. For instance, low THRβ expressing tumors were associated with poor outcome in triple negative BC [24]. Lack of nuclear THRβ1 staining was reported in early stage BC and explained not only by loss of heterozygosity, but also by THRβ1 promotor hypermethylation [25]. More recently, a study in early BC demonstrated that THRβ1 expression is associated with long survival and is an independent prognosis marker [26].

BC signaling and progression is also influenced by the complex interplay between NR, their transcriptional coactivators, and corepressors that also have prognostic significance [27]. Among the transcriptional coregulators, RIP140 (receptor-interacting protein of 140 kDa) and LCoR (ligand-dependent corepressor) play major roles in BC cell proliferation [7]. Moreover, we recently analyzed RIP140 and LCoR expression at the protein level in BC biopsies, showing that expression of these two proteins was highly correlated in more than 80% of tumors and that cytoplasmic RIP140 expression was significantly correlated with a poor patient survival [9]. By sharing the same heterodimerization partner and/or coregulators, other NRs such as peroxisome proliferator-activated receptor γ (PPARγ) or vitamin D receptor (VDR) [6,28] may indirectly impact THR signaling. 

While THRβ1 clearly appears to be a key player in BC carcinogenesis, the importance of its subcellular localization remained to be elucidated. Therefore, purpose of this study was to analyze the nuclear and cytoplasmic localization of THRβ1 in a well-defined cohort of 274 primary BC patients, and to correlate the results with clinicopathological parameters and clinical outcome.

## 2. Results

### 2.1. THRβ1 Expression in Primary Breast Cancers

The total cohort consisted of 274 samples from 271 primary BC patients (Table 1). Approval by the Ethical Committee of the Medical Faculty, Germany had been granted and informed consent was obtained from all patients, as described in Section 4.1. Median age at initial diagnosis was 57.0 years (range 34.8–94.6 years); median follow-up between first diagnosis and last follow-up was 126 months (range 4–153 months). During this period, 39 (14.2%) and 54 (19.7%) cases experienced either local recurrence or distant metastases, respectively; 15 experienced both (5.7%); and 75 (27.4%) women died.

Expression of THRβ1 was analyzed by immunohistochemistry (IHC), using immunoreactive scores (IRS) as described in Material and Methods. Distribution of staining intensities and percentages of stained cells are presented in Appendix A (panels A and B). THRβ1 was widely expressed and detected in 67.3% of the samples with predominantly nuclear location. Cytoplasmic staining also occurred and was quite strong in some cases. Distribution of IRS obtained either for nuclear (C) or cytoplasmic (D) THRβ1 staining (*n* = 263 tumors stained) is presented in Appendix A. It is noteworthy that, for cytoplasmic THRβ1 staining, the highest IRS was 8. This was observed for only two patients (exemplified in Figure 1A, enlarged in B); next to these two cases, 4 was the maximum IRS observed. Consequently, panel C of Figure 1 shows one of the high cytoplasmic THRβ1 IRS (IRS 4). In Figure 1, THRβ1 staining is illustrated for four patients with examples of absent or high expression, and the respective nucleo–cytoplasmic IRS ratio. For extreme nuclear–cytoplasmic ratios (i.e., 0:0 and 12:8), enlarged photos are added (panels B and F).

THRβ1 distribution was then analyzed both in nucleus and in cytoplasm, and total expression (sum of nuclear and cytoplasmic IRS) was calculated (Table 2). Nuclear staining was significantly stronger than the cytoplasmic one (*p* < 0.05), although both means were quite low (1.41 and 1.30, respectively). Nuclear THRβ1 staining was present in 60.5% of the tumors, and cytoplasmic THRβ1 in 43.3%. Interestingly, nuclear and cytoplasmic THRβ1 was significantly and positively correlated with each other (r = 0.440 *p* < 0.01 using Spearman–Rho test).

Distribution of tumors with negative or positive nuclear, or cytoplasmic, THRβ1 staining was analyzed for all 263 tumors stained (Appendix A). It appeared that almost one-third of the tumors were either negative (32.7%) or positive (36.5%) for both nuclear and cytoplasmic THRβ1 localizations. Regarding the nucleo–cytoplasmic ratio, 115 tumors (43.7%) had a ratio of 1, 80 tumors (30.4%) had a ratio greater than 1 (i.e., more expression in the nuclear compartment), and 68 tumors (25.9%) a ratio less than 1 (i.e., more expression in cytoplasm).

### 2.2. Correlation with Nuclear Receptor and Related Coregulators

Using pairwise analysis, we first analyzed the correlation of THRβ1 expression with expression of other NR and coregulators (Table 3) with previously reported expression data [6,28,29,30]. We observed that both nuclear and cytoplasmic expression of THRβ1 was strongly positively correlated with expression of its heterodimerization partner RXR. We also searched for correlation with other NRs, namely, ER, PR, PPARγ, and VDR, and the coregulators LCoR and RIP140. We found no correlation with ER and PR, but saw a strong correlation between nuclear and cytoplasmic THRβ1 and PPARγ. In contrast, only cytoplasmic THRβ1 was significantly correlated with VDR. 

We previously investigated the expression of two NR transcriptional coregulators, namely RIP140 and LCoR, and demonstrated that their sub-cellular localization may define their association with BC aggressiveness and survival [9]. Regarding THRβ1, only cytoplasmic expression of the transcriptional coregulator LCoR was positively correlated with cytoplasmic THRβ1. No significant association was observed with RIP140.

### 2.3. Correlation with Clinicopathological Parameters

Correlations between THRβ1 expression and known clinicopathological characteristics, besides ER and PR, were also analyzed. CD133, a widely used marker for isolating cancer stem cells [31,32], and N-cadherin, a well-known marker for epithelial-to-mesenchymal transition [33], are associated with BC aggressiveness; we previously reported quantification of their expression in the same BC cohort [34]. As shown in Table 4, both nuclear and cytoplasmic THRβ1 expression were significantly and positively correlated with CD133 and N-cadherin (NCAD). Nonetheless, only cytoplasmic THRβ1 expression was positively correlated with proliferation marker Ki67 and HER2, but negatively with tumor size. No further significant correlation between the clinicopathological characteristics mentioned in Table 1 and THRβ1 expression was found.

### 2.4. Correlation between THR Expression and Patient Outcome

In order to analyze the correlation between THRβ1 and patient outcome, we performed Kaplan–Meier analyses (Figure 2). Instead of the simple negative/positive cut-off (Table 2), we determined optimal IRS cut-off values for overall survival (OS) using receiver operating characteristic (ROC)-curve analysis, thus allowing maximum difference between sensitivity and specificity parameters. We then divided the tumors in low or high expressing subgroups for all survival analyses.

As shown in Figure 2A,B, neither nuclear nor cytoplasmic THRβ1 had any significant correlation with relapse-free survival (RFS), for various IRS cut-off values tested. Nonetheless, there was an opposite trend regarding RFS; nuclear THRβ1 expression was related to poor outcome and cytoplasmic expression to favourable outcome. Analyzing OS (Figure 2D,E), we found that the nuclear THRβ1 was significantly correlated with poor outcome, while cytoplasmic THRβ1 was significantly correlated with favourable outcome (*p* = 0.038 and 0.015, respectively). Analyzing total THRβ1 expression of (sum of nuclear and cytoplasmic IRS), no correlation with RFS (Figure 2C) nor OS (Figure 2F) was found.

To determine the specificity of this result (opposite correlations of nuclear and cytoplasmic THRβ1 expression with OS), we performed the identical analyses with another THR, namely THRα2, in the same cohort (staining in Appendix A; distribution in Appendix A). Analyzing OS according to nuclear and cytoplasmic THRα2 expression (Appendix A), we could demonstrate, as expected, a significant beneficial effect of nuclear THRα2 on OS (panel A). Yet, cytoplasmic THRα2 expression did not have any significant correlation with OS (panel B). These additional analyses suggest that the opposite impact on outcome observed for cytoplasmic and nuclear THRβ1 expression may not be true for all THR isoforms.

### 2.5. Nuclear and Cytoplasmic THRβ1 Expression as Independent Prognostic Parameters for OS

Finally, we performed multivariate analyses using the Cox regression model with cytoplasmic and nuclear THRβ1 expression and four relevant clinicopathological features (age at time of diagnosis, tumor size, ER-, and HER2 status). As shown in Table 5, we found that age, tumor size, and ER were independent prognostic markers for OS. As expected, the cytoplasmic form of THRβ1 expression appeared to be an independent prognostic marker for OS with a hazard ratio of 0.545, confirming its correlation with favorable outcome. Interestingly, nuclear THRβ1 expression was shown as an independent prognostic marker for poor OS; with a hazard ratio of 2.860 indicating a higher risk of death for patients whose tumors express high levels of nuclear THRβ1.

## 3. Discussion

The aim of this study was to characterize THRβ1 expression in a wide range of primary BC tissues, taking into account its intracellular expression, and to correlate the results with clinicopathological parameters and patient outcome.

Our study confirmed that THRβ1 is expressed with a predominantly nuclear location, as previously described for most THR isoforms. Nonetheless, our results also demonstrate cytoplasmic localization of THRβ1 in BC. THs are able to modulate gene expression by binding to THRα either in the cytoplasm or in the nucleus of the cells [35]. It is also known that THR can be present not only in the nucleus, but also in the cytoplasm and in the mitochondria [12]. T3 can also be associated to plasma membrane structural α5β3 integrin, thereby regulating cell–cell and cell–extracellular matrix interactions and changing the morphology of BC cells [36]. Our results are supported by a previous study reporting cytoplasmic expression of THRβ1. In a large cohort of early BC patients, THRβ1 expression was predominantly found in the cytoplasm [26]. In most studies, however, including ours, THRβ1 expression is predominantly nuclear. We are aware that the different antibodies used in each study may explain substantial differences in expression. Nonetheless, in another study, THRβ was described as being expressed in nuclei of proliferative cells, in in situ carcinoma, and in the cytoplasm in normal breast and in infiltrative BC cells [37].

The second major observation provided by our study is that nuclear and cytoplasmic forms of THRβ1 may exhibit opposite roles in breast tumorigenesis. Indeed, considering the correlation with patient survival (Figure 2), cytoplasmic expression consistently behaved opposite to nuclear expression. These correlations are strengthened by the fact that nuclear THRβ1 is an independent prognostic marker for poor outcome in multivariate analysis, whereas cytoplasmic THRβ1 is an independent prognostic marker for favorable outcome (Table 5). The only other study that took the subcellular localization of THRβ1 expression into account (*n* = 796) [26] focused solely on cytoplasmic THRβ1, but did not consider nuclear expression. It should be noted that, in our study, both nuclear and cytoplasmic THRβ1 expression correlate with the heterodimerization partner, RXR, but only cytoplasmic THRβ1 correlates with VDR and cytoplasmic LCoR. Consequently, mere analysis of nuclear THRβ1expression, although this is the predominant expression, does not allow a complete understanding of the relevance of both expression types. Considering the subcellular THRβ1 localization seems to be essential for further analysis of its impact on patient outcome. A recent in vitro study suggested a novel role of THRβ, namely THRβ1, in the biology of cancer stem cells that could explain its action as a tumor suppressor in BC [38]. In our study, both nuclear and cytoplasmic THRβ1 strongly correlate positively with CD133 and N-cadherin, without any differential effect according to their subcellular location.

Concerning the link with ER, the study by Jerzak et al. [26] reported a correlation of cytoplasmic THRβ1 with favorable outcome only in ER-positive BC. Although we did not see a significant correlation between ER expression and nuclear or cytoplasmic THRβ1 expression (Table 3), we confirmed that cytoplasmic THRβ1 expression was correlated with good outcome in ER-positive tumors (*p* = 0.021), but not in ER-negative ones (*p* = 0.161) (Appendix A). Consequently, we demonstrated that cytoplasmic THRβ1 expression was also correlated with good outcome in luminal tumors (*p* = 0.035), but not in non-luminal ones (*p* = 0.142) (Appendix A). Yet, when we stratified our cohort according to ER expression, nuclear THRβ1 was no longer correlated with OS in either subgroup (data not shown). Further investigations are needed to define the link between cytoplasmic THRβ1 and estrogen signaling in BC cells at the molecular level.

Our results also suggest that the differential impact on outcome depending on nuclear or cytoplasmic THRβ1 localization is not a common feature for all THRs. In the present study, we also analyzed THRα2 expression. Previously, we had demonstrated that nuclear THRα2 expression tends to be an independent and favorable prognostic marker for survival in a small cohort of 82 invasive BC cases [39]. This was confirmed in another cohort of 130 invasive BC samples, where THRα2 (nuclear and cytoplasmic) negatively correlated with HER2 status, and positively with ER/PR and favorable OS [40]. In the present work, we confirmed that nuclear THRα2 was significantly correlated with a favorable prognosis. Interestingly, we did not find any inverse correlation of cytoplasmic THRα2 expression with OS (Appendix A). Taken together, our data suggest a specific role of each subcellular expression only for THRβ1.

In summary, the present study confirms the complexity of the links between subcellular localization of the THRβ1 protein and its association with patient outcome. To our knowledge, it is the only study supporting the fact that the nuclear form of THRβ1, probably acting as a classical ligand-dependent transcription factor, may have tumor-promoting effects in BC. Our results emphasize the importance of more precise investigations of the subcellular localization of THRs in order to define their impact as potential biomarkers in breast cancer.

## 4. Materials and Methods

### 4.1. Patient Cohort

A total of 274 formalin-fixed paraffin-embedded primary BC tissues were collected from 271 patients (3 of them with bilateral BC) who underwent surgery between 2000 and 2002 at the Department of Obstetrics and Gynecology of the Ludwig-Maximilians-University Munich, Germany. All patient data and clinical information from the Munich Cancer Registry were fully anonymized and encoded for statistical analysis. Research was approved by the Ethical Committee of the Medical Faculty, Ludwig-Maximilian-University (LMU), Munich, Germany (approval number 048-08; 18 March 2008) and informed consent was obtained from all patients. Union for International Cancer Control (UICC) TNM classification was performed to evaluate the size and extent of the primary tumor (pT), lymph node involvement (pN), and distant metastasis (M). Tumor grade was determined by an experienced pathologist (Dr D. Mayr) of the LMU Department of Pathology, according to a modification of Elston and Ellis grading proposed by Bloom and Richardson [41]. ER, PR, HER2, Ki67, and histological status were determined by an experienced pathologist (LMU Department of Pathology), as described below. HER2 2+ scores were further evaluated through fluorescence in situ hybridization (FISH) testing.

### 4.2. Immunohistochemistry (IHC)

Expression of ERα and PR was determined at diagnosis in all BC samples of this cohort at the LMU Department of Pathology, Germany. ERα and PR expression were evaluated by IHC, as previously described [6,30]. Samples showing nuclear staining in more than 10% of tumor cells were considered as hormone receptor-positive, in agreement with the guidelines at the time of analysis (2000–2002). HER2 expression was later analyzed using an automated staining system (Ventana; Roche, Mannheim, Germany), according to the manufacturer’s instructions. Ki67 was stained using an anti-Ki67 monoclonal antibody (Dako, Hamburg, Germany) at a dilution of 1:150 on a VENTANA^®^-Benchmark Unit (Roche, Mannheim, Germany) as previously described [28]. The Ki67 cut-off used to differentiate luminal A from luminal B tumors (all HER2 negative) was 14%, as this was commonly used at the time of the analysis, although 20% is now preferred [42]. We performed paired-analysis, and used data on N-cadherin and CD133 expression in these BC samples extracted from a previously published study [34], as well as RXR, VDR and PPARγ [28,30], and RIP140 and LCoR [9]. For THRα and THRβ1 analysis by IHC, samples were processed as previously described [9,10,30,34,39,43]. All sections were first cut and prepared from paraffin-embedded BC samples using standard protocols. Phosphate buffered saline (PBS) was used for all washes and sections were incubated in blocking solution (ZytoChem Plus HRP Polymer System Kit, ZYTOMED Systems GmbH, Berlin, Germany) before incubation with primary antibodies. All primary antibodies were rabbit IgG polyclonal: anti- both THRα1 and THRα2 (immunogen being a synthetic peptide corresponding to a region within internal sequence amino acids 246–295 of human thyroid hormone receptor alpha 1 and 2, Abcam, ab 105003, Cambridge, UK) and anti-THRβ1 (immunogen being a synthetic peptide within amino acids 1–100 of the N-terminus of human TR-beta protein, Zytomed, 520-4074, Berlin, Germany). Isoform specific antibodies against THRα2 have been used for Appendix A, namely a monoclonal mouse one against the N-terminus region of THRα2 (MCA 2842, AbD Serotec, Oxford, UK).

After incubation with a biotinylated secondary anti-rabbit or anti-mouse IgG antibody, and with the associated avidin–biotin–peroxidase complex (both Vectastain Elite ABC Kit; Vector Laboratories, Burlingame, CA, USA), visualization was performed with substrate and chromogen 3, 3-diamino-benzidine (DAB; Dako, Glostrup, Denmark). Negative and positive controls were used to assess the specificity of the immunoreactions. Negative controls (colored in blue) were performed in BC tissue by replacement of the primary antibodies by species-specific (rabbit/mouse) isotype control antibodies (Dako, Glostrup, Denmark). Appropriate positive controls (placenta samples) were included in each experiment. Sections were counterstained with acidic hematoxylin, dehydrated, and immediately mounted with Eukitt (Merck, Darmstadt, Germany) before manual analysis with a Diaplan light microscope (Leitz, Wetzlar, Germany) with 25× magnification. Pictures were obtained with a digital Charged Coupled Device (CCD) camera system (JVC, Tokyo, Japan). All slides were analyzed by two or three independent examiners.

### 4.3. Immunoreactive Score (IRS)

Expression of THRβ1 and THRα2 was assessed according to IRS, determined by evaluating the proportion of positive tumor cells, scored as 0 (no staining), 1 (≤10% of stained cells), 2 (11%–50% of stained cells), 3 (51%–80% of stained cells), and 4 (≥81% of stained cells); as well as their staining intensity, graded as 0 (negative), 1 (weak), 2 (moderate), and 3 (strong) (IRS = percentage score × intensity score), as presented in Appendix A (panels A and B). Thus, IRS values range from 0 to 12. As previously described for LCoR and RIP140 [9] and for AhR [10], cytoplasmic and nuclear staining of THRβ1 and THRα2 were evaluated in parallel, with a separate determination of cytoplasmic IRS and nuclear IRS. For all other markers, staining and IRS were determined in the whole cells, without differentiation of nuclear and cytoplasmic staining. A total of one hundred cells (three spots with around thirty cells each) was analyzed for each sample and the IRS corresponded to the mean of the IRS determined on the three spots. The intensity and distribution pattern of the immunochemical staining reaction was evaluated by two independent blinded observers. In five cases (2% of the total), the evaluation of the two observers differed. These cases were re-evaluated by both observers together. After the re-evaluation, both observers agreed on the result. The concordance before the re-evaluation was 98.0%.

### 4.4. Statistical and Survival Analysis

Statistical analyses were performed using software package used for interactive, or batched, statistical analysis (SPSS) 24 (IBMSPSS Statistics, IBM Corp., Armonk, NY, USA). For all analyses, *p* values below 0.05 (*), 0.01 (**), or 0.001 (***) were considered statistically significant. Differences in Table 2 were calculated using mean or percentage bilateral analysis. Receiver operating characteristic (ROC) curve analyses were performed to calculate the optimal cut-off values between low and high THRβ1 and THRα2 expressions, based upon the maximal differences of sensitivity and specificity. The threshold determined regarding OS was an IRS ≥ 2.5 for nuclear THRβ1, ≥1.5 for cytoplasmic THRβ1, ≥1.5 for cytoplasmic THRα2, and ≥0.5 for nuclear THRα2.

Correlation analyses presented in Table 3 and Table 4 were performed by calculating the Spearman–Rho correlation coefficient (*p* values of Spearman–Rho test presented), using pairwise analysis. Survival times were compared by Kaplan–Meier graphics and differences in RFS and OS were tested for significance using the chi-square statistics of the log rank test. Data were assumed to be statistically significant in the case of *p*-value <0.05 or <0.01. Kaplan–Meier curves and estimates were then provided for each group and each marker. The *p* value and the number of patients analyzed in each group are given for each chart.

The multivariable analysis for outcome (OS) presented in Table 5 was performed using the Cox regression model and included nuclear and cytoplasmic of THRβ1 expressions and relevant clinicopathological characteristics as independent variables. Variables were selected based on theoretical considerations and forced into the model. *p* values and hazard ratios were indicated, knowing that the hazard ratios of covariates are interpretable as multiplicative effects on the hazard, and holding the other covariates constant.

## 5. Conclusions

Although THRβ1 was predominantly expressed in tumor cell nuclei in our primary BC cohort, cytoplasmic expression was also detected; its correlation with patient survival was inverse to that of nuclear THRβ1. Our results demonstrate that THRβ1 may have different roles in tumorigenesis according to its subcellular localization. A major conclusion is also that THR, particularly nuclear THRβ1, can exhibit tumor-promoting activities in the mammary gland, as demonstrated by its independent prognostic value.

## Figures and Tables

**Figure 1 ijms-21-00330-f001:**
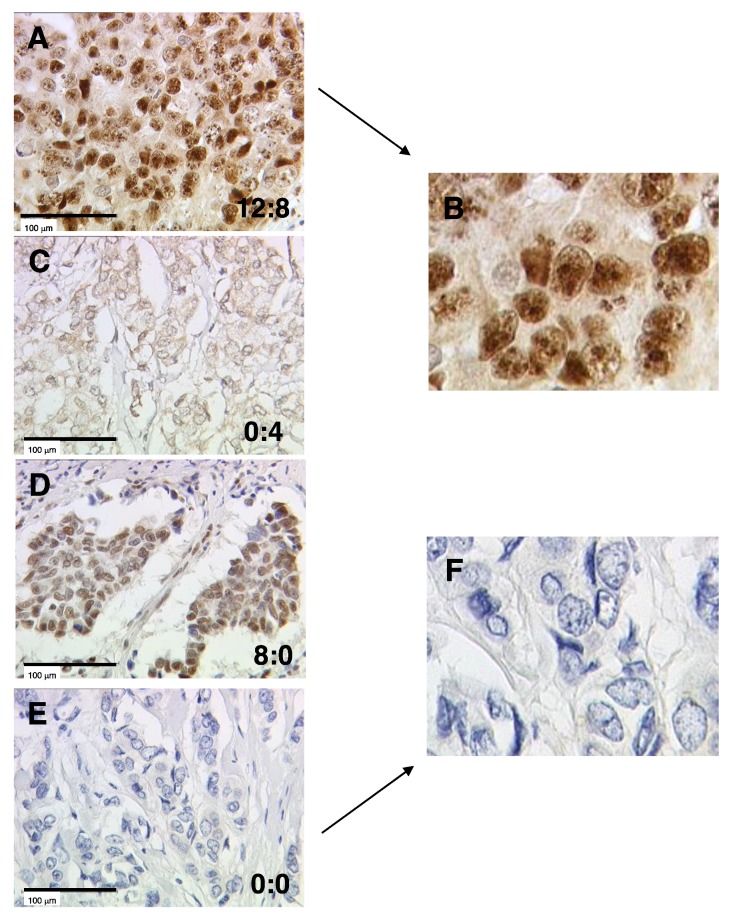
Immunohistochemical staining of thyroid hormone receptor β1 (THRβ1) in breast cancer samples. THRβ1 staining is illustrated for four patients (**A**,**C**–**E**) with examples of absent or high expression. Samples (**A**,**E**) are enlarged in panels (**B**,**F**), respectively. Nucleo–cytoplasmic IRS (immunoreactive score) ratios are indicated in each photomicrograph (25× magnification) and the scale bar equals 100 μm.

**Figure 2 ijms-21-00330-f002:**
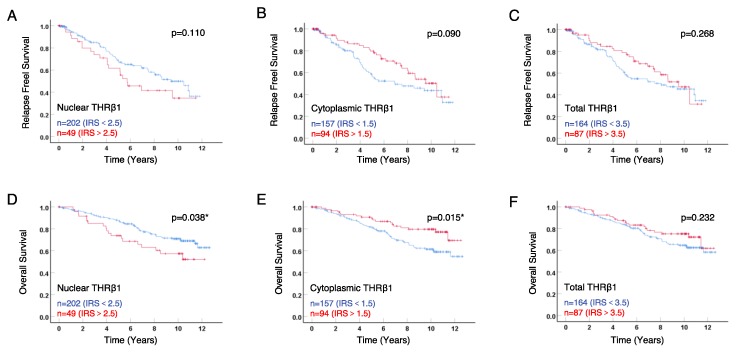
Kaplan–Meier survival analyses according to nuclear, cytoplasmic, and total THRβ1 expression. Relapse free survival (RFS) (**A**–**C**) and overall survival (OS) (**D**–**F**) curves are presented according to THRβ1 expression—either nuclear (**A**,**D**), cytoplasmic (**B**,**E**), or total (**C**,**F**) expression. Optimal IRS cut-off values and number of cases for each group are indicated in each graph. Statistical significance is shown as *p*-value from log-rank test (* *p* < 0.05).

**Table 1 ijms-21-00330-t001:** Clinical and pathological characteristics of all patients.

Clinical and Pathological Characteristics ^a^		%
Median age (years, *n* = 274) ^b^	57.00	range 34.79–94.62
Median follow up (months, *n* = 274) ^b^	126	range 4–153
Histology ^c^ (*n* = 260)		
No Special Type (NST)	139	53.46%
NST with DCIS	74	28.46%
Other invasive	47	18.08%
ER status (*n* = 272)		
Positive	219	80.51%
Negative	53	19.49%
PR status (*n* = 272)		
Positive	160	58.82%
Negative	112	41.18%
HER2 status (*n* = 273)		
Positive	27	9.89%
Negative	246	90.11%
Molecular subtype (*n* = 273)		
Luminal A (Ki-67 ≤ 14%)	152	55.68%
Luminal B (Ki-67 > 14%)	60	21.98%
HER2 positive luminal	20	7.33%
HER2 positive non luminal	7	2.56%
Triple negative	34	12.45%
Grade (*n* = 152)		
I	13	8.55%
II	95	62.50%
III	44	28.95%
Tumor size (*n* = 261)		
pT1	169	64.75%
pT2	78	29.89%
pT3	4	1.53%
pT4	10	3.83%
Lymph node metastasis (*n* = 256)		
Yes	112	43.75%
No	144	56.25%
Distant metastases ^d^ (*n* = 261)		
Yes	54	20.69%
No	207	79.31%
Local recurrence (*n* = 261)		
Yes	39	14.94%
No	222	85.06%

^a^ All information refers to the primary tumor; ^b^ 3 of 271 patients have bilateral primary breast cancer (BC); here, we consider each tumor as an individual one (*n* = 274); ^c^ NST include the formerly called “invasive ductal” and “other” types; ^d^ distant metastasis was detected during the follow-up in 53 patients (1 of them is bilateral BC, so *n* = 54). DCIS, ductal carcinoma in situ; ER, estrogen receptor; PR, progesterone receptor; HER2, human epidermal growth factor receptor 2; Ki67 (also known as MKI67) is a cellular marker for proliferation.

**Table 2 ijms-21-00330-t002:** Distribution of thyroid hormone receptor β1 (THRβ1) expression.

	Nuclear	Cytoplasmic
Mean IRS ± SE	1.41 ± 0.11	1.30 * ± 0.11
Median IRS	1	0
IRS range	0–12	0–8
Number of samples with negative expression **	104 (39.54%)	149 (56.65%)
Number of samples with positive expression **	159 (60.46%)	114 (43.35%)

* Correlations were statistically significant for *p* < 0.05 (*), using Spearman–Rho test using mean bilateral analysis; ** negative defined as immunoreactive score (IRS) = 0, and positive expression as IRS > 0; SE = standard error of means.

**Table 3 ijms-21-00330-t003:** Correlation between THRβ1 expression and nuclear receptors and related coregulators.

	*n*	References	Nuclear	Cytoplasmic
RXR	246	[28,30]	0.256 **	0.186 **
ER	262		0.043	−0.115
PR	262		0.085	−0.014
PPARγ	247	[28,30]	0.315 **	0.247 **
VDR	248	[28]	−0.097	−0.155 *
LCoR	257	[9]		
Nuclear			0.011	−0.060
Cytoplasmic			0.110	0.221 **
RIP140	258	[9]		
Nuclear	262		0.027	−0.046
Cytoplasmic	262		−0.009	0.029

Correlations are statistically significant for *p* < 0.05 (*) or *p* < 0.01 (**), using Spearman–Rho test. RXR, retinoid X receptor; PPARγ, peroxisome proliferator-activated receptor γ; VDR, vitamin D receptor; LCoR, ligand-dependent corepressor; RIP140, receptor interacting protein of 140 kDa.

**Table 4 ijms-21-00330-t004:** Correlation between THRβ1 expression and clinicopathological markers. HER2, human epidermal growth factor receptor 2; NCAD, N-cadherin.

	*n*	Nuclear	Cytoplasmic
pT	251	−0.023	−0.151 *
pN	247	0.044	−0.066
Grade	145	0.128	0.101
HER2 status	262	0.080	0.131 *
Triple negative	263	−0.052	0.031
Ki67	204	0.089	0.225 **
CD133	240	0.183 **	0.178 **
NCAD	244	0.342 **	0.327 **

Correlations are statistically significant for *p* < 0.05 (*) or *p* < 0.01 (**), using Spearman–Rho test; CD = cluster of differentiation.

**Table 5 ijms-21-00330-t005:** Multivariate analysis (OS, overall survival) of clinicopathological variables and THRβ1.

Variable	*p*-Value	HR (95% CI)
Age	0.000007 ***	1.042 (1.023–1.061)
pT	0.0000002 ***	3.701 (2.256–6.073)
ER	0.001 **	0.408 (0.242–0.687)
HER2	0.209	1.566 (0.778–3.153)
Cytoplasmic THRβ1	0.048 *	0.545 (0.299–0.995)
Nuclear THRβ1	0.0004 **	2.860 (1.597–5.119)

Hazard ratios (HRs) are indicated with 95% confidence intervals (CIs). Correlations are statistically significant for *p* < 0.05 (*), *p* < 0.01 (**), or *p* < 0.001 (***).

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
