# Peer review of "Cytoplasmic and Nuclear Forms of Thyroid Hormone Receptor β1 Are Inversely Associated with Survival in Primary Breast Cancer"

_ijms, 2020, doi:10.3390/ijms21010330_

Round 1
Reviewer 1 Report
The manuscript by Shao et al. titled 'Cytoplasmic and nuclear forms of thyroid hormone receptor β1 are inversely associated with survival in primary breast cancer.' is informative and may have clinical importance in prognosis of breast cancer patients. Authors have showed that nuclear TRb expression has worse prognosis than cytoplasmic TRb in BC patients and prioritise the treatment as these patients are at high risk. I only have a minor comment as in line 93/94, authors stated that cytoplasmic staining was quite strong in histological section presented in figure 1 A-D, however, sections B,C and D are not showing strong staining at al.
Author Response
Ref: ijms-664373, Shao et al.
Cytoplasmic and nuclear forms of thyroid hormone receptor β1
are inversely associated with survival in primary breast cancer
Response to reviewer’s comments
In the revised manuscript, all changes have been highlighted in yellow
Reviewer 1
Minor comment: in line 93/94, authors stated that cytoplasmic staining was quite strong in histological section presented in figure 1 A-D, however, sections B,C and D are not showing strong staining at al.
Following the reviewer’s comment, we modified Figure 1 by selecting new photos from 4 patients with extreme values of THRb1 nuclear and cytoplasmic IRS. For the extreme nuclear-cytoplasmic ratios (i.e. 0:0 and 12:8), we also added enlarged photos.
It is noteworthy that, for cytoplasmic THRβ1 staining, the highest IRS was 8. This was observed for only two patients (exemplified in Figure 1A, enlarged in B); next to these two cases, 4 was the maximum IRS observed. Consequently, panel C of Figure 1 shows one of the high cytoplasmic THRβ1 IRS (IRS 4).
To make it clearer, the IRS distributions for nuclear (C) and cytoplasmic (D) THRβ1 staining amongst the 263 tumors are now presented in Supplemental Figure A1.
Reviewer 2 Report
The study by Shao et al describes the staining patterns of THRβ1 in a breast cancer cohort, and investigates correlations between the observed THRβ1 patterns and patterns of other NR and clinicopathological data. While I find the data interesting, much more detail and analysis is required. The use of IRS is interesting, but requires more explanation and analysis. Table 3 relies on analysis of THRβ1 combined with several other studies previously published by this group, however they don't provide the original critical data here (even summarised as a key table). I don't believe this is appropriate or suitable, as it limits the readers ability to scrutinise the underlying data/findings used for analysis.
A key point is if the analysis is using the same samples in a paired analysis, if so the findings would be much more impactful and insightful.
The manuscript would benefit from language editing.
Major Points:
The abstract requires more details- such as p values/CI.
There is no statement addressing ethical approval/informed consent.
I don't believe that the unknown samples should be included (Table 1). At the least I believe these should be excluded in analysis of their respective clinicopathological details and the % adjusted to reflect analysis of the known samples only.
I’m not convinced that including Invasive Medullar (n=10) and Invasive mucinous (n=3) types are useful. There are not enough samples to make a significant analysis of these types alone. I suggest removing them.
Figure 1 would benefit from in-panel enlargements to show individual cell staining clearly.
Figure 1: a supplemental image is required to show multiple samples with the indicated staining patterns. Images (multiple independent examples) showing each staining intensity (negative, weak, moderate and strong) is required, either as a panel in the main figure or as a supplemental. What is the proportion of cells in each pattern with the indicated pattern (i.e. what percent of cells need to have strong nuclear staining to be classified as strong, >50%, >70%? When there is a mixed pattern)? More detail is needed.
Table 2 should be described in more detail, both the rationale behind this and how the results were used/relate to the IHC staining classification. How many cells were analysed in each section? How many overall? Was there any normalisation between tumour samples? What was the proportions and range of nuclear and cytoplasmic staining seen in each section? Much more detailed analysis is required. The details for staining in the combined pattern, nuclear and cytoplasmic, is needed in the table.
I would like to see the Figure 1/Table 2 data describe the patterns observed in the whole cohort. How many sections show each of the indicated patterns (nuclear: strong, medium, weak; cytoplasmic: strong, medium, weak ). What percent of sections have mixed patterns, what are the different patterns seen (i.e strong nuclear, strong nuclear/weak cytoplasmic…. ). How do these correlate with the associated clinicopathological details (tumour size, local or metastatic disease, clinical subtype)? i.e. Sentence starting line 102: what are the relevant clinicopathological details in the 60.5 and 43.3%? There is no linking/investigation of the staining pattern to relevant clinicopathological features of these groups.
For all relevant tables, if the n= changes for any variable the n= MUST be given for each variable described. Using a broad range for the table is not acceptable.
Line 103: re the correlation: how is this relevant to any of the key clinicopathological details? I don't see the significance or relevance of this.
Where is the data associated with Table 3? You state it has been previously published, if so these should be presented as analysis of paired observations for each sample- if they are indeed observations from the same samples. In addition I would like to see examples of the both patterns in the same samples. At the least as supplemental images the pattern being associated with THRβ1 staining should be shown (even if published elsewhere- if this data is to be used to support this study. The NR PPARgamma, VDR, RIP140 and LCoR should be described in the introduction if investigated, to provide relevance and context. What is the relevance of exploring co-expression of these particular NR? Do they interact? Regulate each other? Are these NR previously established as effectors/markers of breast cancer progression?
How is CD133 relevant in this context? Are the authors suggesting that the THRβ1 positive cells are CSC?
For Figure 2, how was the “optimal IRS” value calculated? This is key data which should be given (give the ROC-curve analysis data as supplemental data to allow the reader to evaluate your choice of cut off).
How were the now only two groups (high and low expression) determined? How does this relate to the previous high, medium, low and weak categories used? Figure 2 should include analysis of the cytoplasmic and nuclear THRβ1 staining cells?
How were the parameters chosen for Table 5? Why were PR, grade, subtype and other NR not analysed (as previously described earlier in the paper)? This is incomplete.
The primary data for THRalpha2 expression must be shown (as in Figure 1), preferentially with relevant analysis. Either provide the full data set for evaluation, or remove this.
I believe the second major conclusion in the discussion greatly overstates and over-interprets your observational data. No basic data is presented to even show what percent of breast cancer samples had cytoplasmic staining (let alone by subtype)?
Minor Points:
Line 44: I would add: some tumours develop….
Line 51: RXR not defined.
Sentence ending the start of Line 56- references required.
Sentence starting on Line 56: all 4 pathways should be described.
Lines 62/63: define T3 (triiodothyronine). That whole sentence needs rewriting for clarity.
Figure 1 legend: IRS (immunoreactive score) is not defined.
Figure 2: The numbers of patients at year should be displayed on the x axis.
Author Response
Ref: ijms-664373, Shao et al.
Cytoplasmic and nuclear forms of thyroid hormone receptor β1
are inversely associated with survival in primary breast cancer
Response to reviewer’s comments
In the revised manuscript, all changes have been highlighted in yellow
Reviewer 2
Reviewer 2 find the data interesting, but is asking for much more detail and analysis is required. He/she is asking for more explanation and analysis about the IRS use, and eventually a key table about the several other studies previously published by us.
We agree with Reviewer 2 that more analysis strongly improved the manuscript and worked in this direction. As detailed below, we explained in more depth and analyzed further the IRS, the cut-off determinations, and added more information on the results already published by our laboratory on the same cohort of patients. Altogether, 2 figures and 2 tables have been added as Supplemental data, and 3 figures specifically to the Reviewer, presented at the very end of this document.
A key point is if the analysis is using the same samples in a paired analysis, if so the findings would be much more impactful and insightful.
Indeed, the correlation analyses performed using the SPSS software were pairwise and not list wise. This is now mentioned in the revised manuscript both in the Results and in the Materials & Methods.
The manuscript would benefit from language editing.
We performed many changes in order to improve the text of the revised manuscript, and the whole manuscript has been finally carefully edited by Prof. Harbeck, co-author and English fluent scientist.
Major Points:
Comment #1: The abstract requires more details- such as p values/CI.
We now give the p values of the most significant results mentioned in the abstract.
Comment #2: There is no statement addressing ethical approval/informed consent.
The ethical Committee approval and informed consent from patients were described in the patient cohort description (Paragraph 4.1 of the Material and Methods). We have emphasized this point in the revised manuscript at the very beginning of the Results section.
Comment #3: I don't believe that the unknown samples should be included (Table 1). At the least I believe these should be excluded in analysis of their respective clinicopathological details and the % adjusted to reflect analysis of the known samples only.
We agree with Reviewer 2 and recalculated all percentages in Table 1 of the revised manuscript, with the mention of the exact number of cases for each parameter.
Comment #4: I’m not convinced that including Invasive Medullar (n=10) and Invasive mucinous (n=3) types are useful. There are not enough samples to make a significant analysis of these types alone. I suggest removing them.
That has been corrected in Table 1and all invasive cases are now grouped in one unique category.
Comment #5: Figure 1 would benefit from in-panel enlargements to show individual cell staining clearly.
We have significantly modified Figure 1 by selecting new photos from 4 patients with extreme values of THRb1 nuclear and cytoplasmic IRS. For the extreme 0:0 and 12:8 nuclear-cytoplasmic ratios, we have added picture enlargement, as suggested by Reviewer 2.
Comment #6: Figure 1: a supplemental image is required to show multiple samples with the indicated staining patterns. Images (multiple independent examples) showing each staining intensity (negative, weak, moderate and strong) is required, either as a panel in the main figure or as a supplemental. What is the proportion of cells in each pattern with the indicated pattern (i.e. what percent of cells need to have strong nuclear staining to be classified as strong, >50%, >70%? When there is a mixed pattern)? More detail is needed
In the revised manuscript, more details are given about the IRS determination (in paragraph 2.1 of Results and in paragraph 4.3 of Materials and Methods). Moreover, we have added Supplemental Figure A1 that presents the distribution of intensity (A) and percentages (B) of nuclear and cytoplasmic THRβ1 stainings (the two parameters which allowed the IRS calculation).
As mentioned in our reply to comment #5, Figure 1 and related text have been modified to better illustrate the extreme IRS values. It is noteworthy that, for cytoplasmic THRβ1 staining, the highest IRS was 8. This was observed for only two patients (exemplified in Figure 1A, enlarged in B); next to these two cases, 4 was the maximum IRS observed. Consequently, panel C of Figure 1 shows one of the high cytoplasmic THRβ1 IRS (IRS 4). Indeed, the mean IRS values for nuclear and cytoplasmic THRβ1 stainings are quite low, 1.41 and 1.30 respectively.
To make it clearer, the distributions of the IRS obtained either for nuclear (C) or cytoplasmic (D) THRβ1 stainings for the 263 stained tumors are presented in the new manuscript in the Supplemental Figure A1.
Comment #7: Table 2 should be described in more detail, both the rationale behind this and how the results were used/relate to the IHC staining classification. How many cells were analysed in each section? How many overall? Was there any normalisation between tumour samples? What was the proportions and range of nuclear and cytoplasmic staining seen in each section? Much more detailed analysis is required. The details for staining in the combined pattern, nuclear and cytoplasmic, is needed in the table.
I would like to see the Figure 1/Table 2 data describe the patterns observed in the whole cohort. How many sections show each of the indicated patterns (nuclear: strong, medium, weak; cytoplasmic: strong, medium, weak ). What percent of sections have mixed patterns, what are the different patterns seen (i.e strong nuclear, strong nuclear/weak cytoplasmic…. )..
We agree that the technical aspects of the protocols were not detailed in the manuscript submitted. The revised manuscript has been improved, with the following additions :
The following text about IRS determination has been added in paragraph 4.3 of the Material and Methods: “A total of hundred cells (3 spots with around thirty cells each) was analyzed for each sample and the IRS corresponded to the mean of the IRS determined on the three spots. Intensity and distribution pattern of the immunochemical staining reaction were evaluated by two independent blinded observers. In 5 cases (2 % of the total), the evaluation of the two observers differed. These cases were re-evaluated by both observers together. After the re-evaluation, both observers agreed on the result. The concordance before the re-evaluation was 98.0%.”
The Supplemental Table 1 now gives the distribution of tumors with negative or positive nuclear, or cytoplasmic, THRβ1. The following text has been added in paragraph 2.1 of the Results section in the revised manuscript, after the description of Table 2.
“It appeared that almost one third of tumors was either negative (32.7%) or positive (36.5% ) for both nuclear and cytoplasmic THRβ1 localizations. Regarding the nucleo:cytoplasmic ratio, 115 tumors (43.7%) had a ratio of 1, 80 tumors (30.4%) had a ratio greater than 1 (i.e. more expression in the nuclear compartment) and 68 tumors (25.9%) a ratio less than 1 (i.e. more expression in cytoplasm).”
Comment #8: How do these correlate with the associated clinicopathological details (tumour size, local or metastatic disease, clinical subtype)? i.e. Sentence starting line 102: what are the relevant clinicopathological details in the 60.5 and 43.3%? There is no linking/investigation of the staining pattern to relevant clinicopathological features of these groups.
We agree with Reviewer 2 that the paragraph describing Table 2 was not clear. In the revised manuscript, we first describe, as explained above in the answer to Comment #6, the Supplemental Figure A1 with the exact distribution of nuclear (C) and cytoplasmic (D) THRβ1 stainings, and then clarified the fact that 60.5 and 43.3% are the percentage of tumors expressing nuclear and cytoplasmic THRβ1 (this is now also mentioned in Table 2 itself).
This paragraph had no link with the clinicopathological characteristics. Those are studied in paragraph 2.3., where we made clear that the characteristics described in Table 1 not presented in the correlation results of Table 4, showed no significant correlation with neither nuclear nor cytoplasmic THRβ1: “No further significant correlation between clinicopathological characteristics mentioned in Table 1 and THRβ1 expression was found.”
Comment #9: For all relevant tables, if the n= changes for any variable the n= MUST be given for each variable described. Using a broad range for the table is not acceptable.
The n values have been specified for each parameter in all correlation tables.
Comment #10: Line 103: re the correlation: how is this relevant to any of the key clinicopathological details? I don't see the significance or relevance of this.
See answer to Comment #8.
Comment #11: Where is the data associated with Table 3? You state it has been previously published, if so these should be presented as analysis of paired observations for each sample- if they are indeed observations from the same samples. In addition I would like to see examples of the both patterns in the same samples. At the least as supplemental images the pattern being associated with THRβ1 staining should be shown (even if published elsewhere- if this data is to be used to support this study. The NR PPARgamma, VDR, RIP140 and LCoR should be described in the introduction if investigated, to provide relevance and context. What is the relevance of exploring co-expression of these particular NR? Do they interact? Regulate each other? Are these NR previously established as effectors/markers of breast cancer progression?
Because we performed pairwise analysis, we could compare the data we obtained specifically for this study (namely THRβ1 and THRa2) and the data from other markers we already published using this cohort. The references associated to the parameters already published by our group and used in this study are now added in Table 3, with mention in Paragraphs 2.2 and 2.3 of the Results, and 4.2 of the Material and Methods (references [9], [28], [30] and [34]).
Regarding the THR partners other than RXR (namely PPARγ and VDR and the coregulators LCoR and RIP140), we now presented them in the Introduction of the revised manuscript, with the related references. The additional Figure 2 at the end of the present document illustrates the 7 different stainings performed for 2 patients, one with all negative stainings (very uncommon, only 2 patients within the whole cohort) and the other one with only positive stainings, in order to illustrate the correlations of Tables 3 and 4. We decided not to add this Figure in the manuscript in order to keep the number of Supplemental data reasonable.
Comment #12: How is CD133 relevant in this context? Are the authors suggesting that the THRβ1 positive cells are CSC?
We now explain in the revised manuscript that CD133 is a widely used marker for isolating cancer stem cell and is associated with BC aggressiveness [31] [32]. Moreover, we previously reported the quantification of CD133 expression in the same cohort [34]. These explanations and references are added in the paragraph 2.3 of the Results of the revised manuscript.
Regarding the possibility for THRβ1 positive cells to be cancer stem cells, a recent in vitro study suggests a possible role of THRβ1 in the biology of cancer stem cells that could explain its action as a tumor suppressor in BC [38]. This reference has been added in the Discussion, although in our cohort, both nuclear and cytoplasmic THRβ1 expression strongly correlate positively with CD133.
Comment #13: For Figure 2, how was the “optimal IRS” value calculated? This is key data which should be given (give the ROC-curve analysis data as supplemental data to allow the reader to evaluate your choice of cut off). How were the now only two groups (high and low expression) determined? How does this relate to the previous high, medium, low and weak categories used? Figure 2 should include analysis of the cytoplasmic and nuclear THRβ1 staining cells?
We agree that the various IRS cut-off used for the different analyses can be confusing. We now clearly state in the revised manuscript that we first considered negative/positive expression of nuclear and cytoplasmic THRβ1 thus corresponding to IRS=0 and IRS>0 (Table 2 and Supplemental Table 1 for descriptive Paragraph 2.1).
Then for all survival analyses from Paragraph 2.4, we used the specific IRS cut-off determined by the ROC-curve analyses. The ROC-curve analysis data are now shown for Reviewer 2 only in the additional Figure 1 presented at the end of this document. This analysis is more detailed in the revised manuscript, in paragraph 2.4 of Results and in paragraph 4.4 of Materials and Methods.
Comment #14: How were the parameters chosen for Table 5? Why were PR, grade, subtype and other NR not analysed (as previously described earlier in the paper)? This is incomplete.
The priority goal of this analysis was to demonstrate if nuclear and/or cytoplasmic THRβ1 could be independent prognosis markers. So we used the well recognized independent prognosis markers and compared THRβ1 expression to them. The choice of the 4 characteristics used was determined by the best results obtained for both nuclear and cytoplasmic THRβ1, compared to other possibilities including more clinicopathological characteristics. For example, only nuclear THRβ1 (but not cytoplasmic THRβ1) remains an independent prognosis marker if the grading, the triple negative or Ki67 status is added. We slightly changed accordingly the first sentence in paragraph 2.5 of Results.
Comment #15: The primary data for THRalpha2 expression must be shown (as in Figure 1), preferentially with relevant analysis. Either provide the full data set for evaluation, or remove this.
The primary data of THRa2 expression are now presented in Supplemental Figure A2 and Supplemental Table 2 and mentioned before analyzing the OS according to THRa2 nuclear and cytoplasmic expression (now Supplemental Figure A3).
Comment #16: I believe the second major conclusion in the discussion greatly overstates and over-interprets your observational data. No basic data is presented to even show what percent of breast cancer samples had cytoplasmic staining (let alone by subtype)?
We agree that this assumption was too strong and moderated the conclusion.
Minor Points:
Comment #17: Line 44: I would add: some tumours develop….
This has been added.
Comment #18: Line 51: RXR not defined.
The role of RXR has been added in the Introduction and the definition was in the “Abbreviation” section
Comment #19: Sentence ending the start of Line 56- references required
The reference [12] from Anyetei-Anum et coll. has been added.
Comment #20: Sentence starting on Line 56: all 4 pathways should be described.
The description of the 4 pathways has been added.
Comment #21: Lines 62/63: define T3 (triiodothyronine). That whole sentence needs rewriting for clarity.
For clarity, T3 has been replaced by TH, already defined in the text of the Introduction, and the sentence has been rephrased.
Comment #22: Figure 1 legend: IRS (immunoreactive score) is not defined
The definition has been added.
Comment #23: Figure 2: The numbers of patients at year should be displayed on the x axis.
Figure 2 has been implemented as requested. We agree that this information is important, but the addition in the 6 panels of Figure 2 make it very hard to read. To keep the clarity of the manuscript, we added it as additional Figure 3 below for Reviewer 2 appreciation, and propose not to replace it in the revised manuscript.
Additional figures for Reviewer only:
Additional Figure 1. ROC-curves and associated data used for cut-off determination of nuclear and cytoplasmic THRβ1 and THRα2 IRS. ROC-curve analysis for OS are represented for either the nuclear (A) or cytoplasmic (B) THRβ1 expression and for either the nuclear (C) or cytoplasmic (D) THRα2 expression. Each IRS cut-off tested is mentioned with the related sensitivity and 1 - specificity, to identify the IRS that allow to reach the maximal difference.
Additional Figure 2. Immunohistochemical staining of 7 correlated parameters in BC samples from 2 patients. THRβ1, THRα2, RXR, PPARg, LCoR, CD133 and NCAD stainings are illustrated for 2 patients with examples of null (Patient 357, top line) or positive (and predominantly high, patient 522, bottom line) expressions. Nucleo:cytoplasmic IRS (immunoreactive score) ratios, or nuclear IRS, are indicated in each photomicrograph (25× magnification) and scale bar equals 100μm.
Additional Figure 3. Kaplan-Meier survival analyses according to nuclear, cytoplasmic, and total THRβ1 expression. Relapse Free Survival (RFS) (A, B and C) and Overall Survival (OS) (D, E and F) curves are presented according to THRβ1 expression – either nuclear (A and D), cytoplasmic (B and E) or total (C and F) expression. Number of events and patients at risk are given every 2 years. Optimal IRS cut-off values and number of cases for each group are indicated in each graph. Statistical significance is shown as p-value from log-rank test (*: p<0.05).
